# Automatic Control of the Middle Route Project for South-to-North Water Transfer Based on Linear Model Predictive Control Algorithm

**Lingzhong Kong [1], Jin Quan [2,*], Qian Yang [3], Peibing Song [1] and Jie Zhu [4]**

1   College of Civil Engineering and Architecture, Zhejiang University, Hangzhou 310058, China;
    lzkong@126.com (L.K.); songpeibing@zju.edu.cn (P.S.)
2   Department of Water Resources, China Institute of Water Resources and Hydropower Research,
    Beijing 100038, China
3   College of Water Conservancy and Hydropower Engineering, Hohai University, Nanjing 210098, China;
    15251704382@163.com
4   College of Architecture and Civil Engineering, Beijing University of Technology, Beijing 100124, China;
    de_lovely666@163.com
*   Correspondence: jeanquan@163.com; Tel.: +86-10-8820-6756

**Abstract:** The application of automatic control to irrigation canals is an important means of improving the efficiency of water delivery. The Middle Route Project (MRP) for South-to-North Water Transfer, the largest water transfer project in China, is currently under manual control. Given the complexity of the MRP, there is an urgent need to adopt some form of automatic control. This paper describes the application of model predictive control (MPC), a popular real time control algorithm particularly suited to the automatic control of multi-pool irrigation water delivery systems, to the MRP using a linear control model. This control system is tested in part of the MRP by means of numerical simulations. The results show that the control system can deal with both known and unknown disturbances, albeit with a degree of resonance in some short pools. However, it takes a long time for the MRP to reach a stable state under the MPC system and the calculation time for the whole MRP network would be too long to satisfy the requirements of real-time control. Suggestions are presented for the construction of an automatic control system for the MRP.

**Keywords:** canal automation; MPC algorithm; MRP; real-time control; hydraulic models

## 1. Introduction

Long-distance water transfer is the most effective and direct means of adjusting the uneven spatial and temporal distribution of water resources, and enables differences in the supply and demand of water resources to be overcome [1]. The control goal of water delivery systems is to provide users with safe and reliable water supply services, requiring flexible adjustments to the flow to meet the needs of users [2]. The stability of the water supply flow is achieved by maintaining water level stability at certain points, which are usually referred to as control points. Normally, these control points are at the downstream end of the pools. Water level control is mainly accomplished by certain control structures, such as control gates or pumping stations, at the ends of the canal pools.

The Middle Route Project (MRP) for South-to-North Water Transfer, which began to deliver water in December 2014, is the largest open canal water transfer project in China. The project delivers water to some large cities in Henan province and Hebei province, along with Tianjin City and Beijing City, the capital of China, and has become the main source of urban water in Tianjin City and Beijing City [3]. The main canal of the MRP is 1273 km long, and has a design flow rate of 350 m$^3$/s on the upstream

side and 50 m$^3$/s on the downstream side. In 2018, the operational flow rate was about 200 m$^3$/s on the upstream side and 40 m$^3$/s on the downstream side. The canal has 63 undershot check gate stations, one pump station, and 97 offtakes to customers. There is no online reservoir and the entire system is divided into 63 segments by the check gates and pump station. A supervisory control and data acquisition system continuously collects data on water levels, gate positions, and offtake flow rates, and controls the offtake flow and check gates remotely. Although the hardware of the MRP is relatively new and in very good condition, the generation of control strategies still depends on manual experience. The effect of this control mode is limited by the experience of the operators. Although operators are able to adapt to unforeseen changes in the behavior of the canal, their performance is limited by fatigue and loss of concentration; to compensate for this, the operators work in shifts. Furthermore, one operator can only operate a limited number of control structures at a time. This control mode results in obvious changes in water level, instability of the offtake flow, and a reduction in the water delivery efficiency of the canal system.

　　Automatic control of the water conveyance and delivery systems could better service the water users in terms of higher water distribution efficiency, greater flexibility, and lower water losses [4]. At present, there are many advanced control methods. The control algorithms for canals are mainly feedforward and feedback control algorithms. The basic idea of the feedforward control algorithm is to optimize the control strategy based on a canal simulation model [5,6]. However, the practicality of this method is poor. For instance, it can only be used for known delivery changes, the information of which should be included in the simulation model; additionally, the control strategies are based on the simulation model, so they are only effective when the results perfectly match the real system [7]. In large-scale, long-distance water transfer projects, especially in the MRP, it is almost impossible for simulation models to achieve high precision. Therefore, feedforward control algorithms cannot be applied alone. Feedback control strategies are based on a control model and control algorithm. By collecting the observed water level information to generate the control scheme, the feedback control algorithm avoids the need for delivery change information and is not strictly dependent on the simulation model. Feedback control algorithms are typically either single-input–single-output (SISO) or multi-input–multi-output (MIMO), depending on the control logic [8,9]. SISO control algorithms mainly take the form of distributed proportional integral (PI) feedback control. In this control logic, each control building controls only one target water level. Cui et al. (2013) [10] tried to use a PI control algorithm with feedforward control logic to achieve real-time control of the MRP. Although the control effect from the simulation results is good, it is mainly determined by feedforward control. Because the feedforward strategy is based on the simulation model, and is then implemented on the simulation model to verify the effect, a good control effect can be achieved without considering the deviation between the simulation model and the actual model, with feedback playing a very small role. In the MIMO control logic, the water level information from all control points is used at the same time, and all control actions are generated simultaneously, which is more suitable for the joint adjustment of multipool canals. The MIMO control algorithm is also called a centralized control algorithm, and typically uses linear quadratic optimal control [11] or model predictive control (MPC). As MPC can deal with various constraint problems and foreseen delivery problems, it is well-suited to real-time automatic control of canal irrigation systems with more complicated working conditions [12–14].

　　However, there have been few studies on the real-time automatic control of MRP, especially using a centralized control algorithm. The main problem is that almost all MRP canal pools are flat; the bottom slope of the MRP canal pools is very small (about $4 \times 10^{-5}$), and the length of canal pools is relatively short, with several canal pools covering only 10 km. Small gate actions will lead to obvious water level oscillations, making it difficult to implement real-time automatic control. Therefore, this paper describes the use of an MPC algorithm to realize real-time automatic control of the MRP. Referring to other projects for the treatment of resonance problems in the control systems of canal pools, a low-pass filter [15–17] is employed to eliminate the resonance as much as possible. In this study, the control algorithm is implemented by establishing a simulation model and then applying a control model to

the simulation model. Different water offtake disturbances are added to the simulation model to test the applicability of the MPC algorithm to the MRP.

## 2. Materials and Methods

### 2.1. Study Area and Test Scenarios

This study considers the Beijing–Shijiazhuang section, which covers the last 13 pools of the MRP. The study area is located in Hebei Province. The starting point of this canal system is near Shijiazhuang City and the end point is near Beijing City (see Figure 1). The study area is regarded as an independent canal system and the flow is non-pressure flow. The reason for choosing this area is that a practical control algorithm for these 13 canal pools is expected to be practical for all pools of the MRP, and the length of the canal pool changes from 10 km to 27 km, so the characteristics of this canal pool cover almost all of those observed on the MRP. The total length of the canal pools in the study area is 227.3 km. There are 11 offtakes and 7 escape gates, some of which are also used as offtakes. The initial flow condition of the canal system is set to be 70% of the design flow, which is the common flow condition. To simplify the description, it is assumed that there is only one offtake in each canal pool. During normal operation, the water level at the control point must not exceed its allowable water level range. It is assumed that the allowable water level range is no more than 0.35 m outside the designed water level bounds. Initially, the water levels at the control points are at the designed target water levels. The downstream and upstream boundaries are set as water level boundaries with a constant water depth of 3 m and 6 m, respectively. The basic parameters and initial flow conditions of the 13 pools are presented in Table 1.

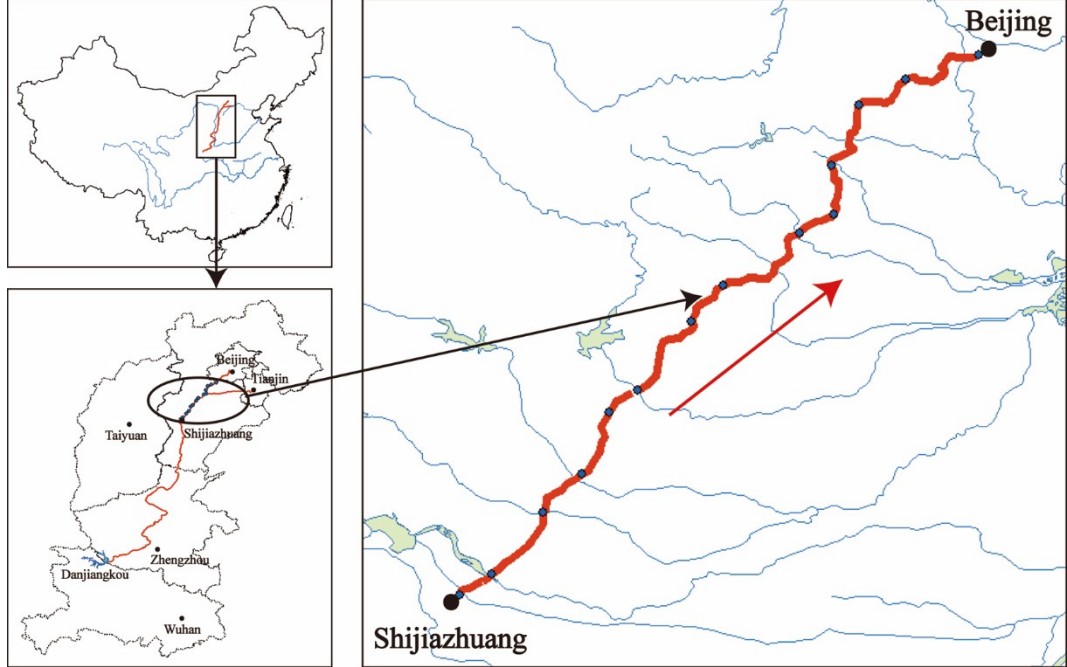

**Figure 1.** Location of the study area.

**Table 1.** Basic parameters and initial flow conditions of each pool.

| Pool | Pool Length (km) | Bottom Width (m) | Side Slope | Slope | Downstream Initial Flows (m³/s) | Offtake Initial Flows (m³/s) | Target Water Depth (m) |
|---|---|---|---|---|---|---|---|
| Heading | | | | | 125.5 | | |
| 1 | 9.4 | 13.2 | 2.5 | $5.58 \times 10^{-5}$ | 125.5 | 0 | 7.2 |
| 2 | 22.0 | 21.5 | 2.5 | $3.89 \times 10^{-5}$ | 118.5 | 7 | 7.16 |
| 3 | 15.2 | 21.5 | 2.5 | $4.05 \times 10^{-5}$ | 113.5 | 5 | 7.23 |
| 4 | 19.5 | 21.5 | 2.5 | $5.56 \times 10^{-5}$ | 109.5 | 4 | 6.77 |
| 5 | 9.2 | 15 | 3 | $3.50 \times 10^{-5}$ | 109.5 | 0 | 6.7 |
| 6 | 25.7 | 21.5 | 2.5 | $3.30 \times 10^{-5}$ | 99.5 | 10 | 4.29 |
| 7 | 13.2 | 21.5 | 2.5 | $6.72 \times 10^{-5}$ | 94.5 | 5 | 7 |
| 8 | 26.6 | 21 | 2 | $9.78 \times 10^{-5}$ | 87 | 7.5 | 4.5 |
| 9 | 9.7 | 22.5 | 2.75 | $3.81 \times 10^{-5}$ | 70 | 17 | 4.5 |
| 10 | 14.9 | 17 | 1 | $6.13 \times 10^{-5}$ | 55 | 15 | 4.21 |
| 11 | 20.8 | 10 | 2 | $5.37 \times 10^{-5}$ | 42 | 13 | 4.19 |
| 12 | 14.7 | 7.5 | 2.5 | $5.13 \times 10^{-5}$ | 42 | 0 | 4.21 |
| 13 | 25.4 | 7.5 | 2.5 | $5.32 \times 10^{-5}$ | 35 | 7 | 3.95 |

Note: Each pool is composed of many sections. Parameters of the bottom width and side slope are approximate numbers.

Four test scenarios are examined to verify the effectiveness of the control system. In scenario 1, the flow at offtake 11 reduces unexpectedly by 5 m³/s. In scenario 2, the flow at offtake 11 reduces by 17 m³/s according to plan, so this change is known in advance. The same water diversion changes are considered in scenario 3, but the offtake change condition is unknown; this scenario provides a direct comparison with scenario 2. Scenario 4 is an emergency condition in which a sudden increase of 35 m³/s occurs at offtake 10.

*2.2. Numerical Simulation of the Study Area*

The simulation of a water delivery canal system mainly includes three parts: canal simulation, control gate simulation, and offtake simulation.

The simulation model of the canal part can be described by the Saint-Venant equations:

$$\begin{cases} \frac{\partial A}{\partial t} + \frac{\partial Q}{\partial x} = q \\ \frac{\partial}{\partial t}\left(\frac{Q}{A}\right) + \frac{\partial}{\partial x}\left(\frac{Q^2}{2A^2}\right) + g\frac{\partial h}{\partial x} + g\left(S_f - S_0\right) = 0 \end{cases} \tag{1}$$

where $x$ and $t$ are the space and time coordinates; $A$ is the wetted area (m²); $Q$ is the flow rate (m³/s); $h$ is the water depth (m); $S_0$ is the canal bottom slope; $g$ is the acceleration of gravity (m/s²); $q$ is the lateral flow rate of the canal for a unit length (m²/s); and $S_f$ is the friction slope, which is defined as

$$S_f = \frac{Q^2 n^2}{A^2 R^{4/3}} \tag{2}$$

with $n$ being the roughness coefficient (s/m$^{1/3}$) and $R$ the hydraulic radius (m), defined by $R = A/P$, where $P$ is the wetted perimeter (m).

The flow equation and water balance equation can be used for control gate simulation. As the flow through the check gates of the MRP is a submerged discharge, the flow equation of the check gates is as follows:

$$Q = C_d l G \sqrt{2g(h_0 - h_S)} \tag{3}$$

where $Q$ is the gate flow rate (m³/s); $G$ is the gate opening (m); $l$ is the gate width (m); $h_0$ is the water depth immediately upstream of the gate (m); $h_S$ is the water depth immediately downstream of the gate (m); and $C_d$ is the discharge coefficient.

The water balance equation of a check gate is as follows:

$$Q_0 = Q_S \tag{4}$$

where $Q_0$ is the flow immediately upstream of the gate (m³/s) and $Q_S$ is the flow immediately downstream of the gate (m³/s).

The water balance equation and the equation describing the relationship between the water depth of the canal section immediately upstream and downstream of the offtake can be used as the simulation equations. As the width of an offtake is generally small, it can be assumed that the water depth of the canal section immediately upstream of the offtake, $h_e$, is the same as that immediately downstream of the offtake, $h_f$, i.e.,

$$h_e = h_f \tag{5}$$

The water balance equation of an offtake is as follows:

$$Q_e = Q_f + Q_i \tag{6}$$

where $Q_e$ is the flow of the canal section immediately upstream of the offtake (m³/s); $Q_f$ is the flow of the canal section immediately downstream of the offtake (m³/s); and $Q_i$ is the flow of the offtake (m³/s), which represents a boundary of the simulation model. The above equations can be used to construct a simulation model of the MRP. The implicit difference scheme [18] is adopted to discretize the above equations and the double sweep method [19] can be used to solve the equations. As real-time control methods generally have good robustness [11,13,20], a model that can describe the characteristics of each pool of the MRP will satisfy the needs of the simulation, so model tuning is not discussed here. The simulation time interval is set to 2 min here.

*2.3. Canal Control Model*

The control model refers to the mathematical model of the process or system to which the control algorithm is directed. Together with the control algorithm, this constitutes a computer-simulated control system. The control model does not use the simulation model, which is computationally intensive and makes implementation of the optimization algorithm difficult. Here, the integral delay (ID) [21] model commonly used in the canal control field is applied to construct the control model. The ID model assumes that the water level deviation of the pool is a linear function of the flow rate change, and the upstream flow change has a delay effect with respect to the water level at the downstream end of the pool. The equation is

$$\frac{\mathrm{d}e}{\mathrm{d}t} = \frac{1}{A_s}\left\{q_{in}(t - t_d) - \left[q_{out}(t) + q_{offtake}(t)\right]\right\} \tag{7}$$

where $e$ is the water level deviation at the downstream end of the pool; $q_{in}(t - t_d)$ is the inflow deviation to the backwater section with delay time $t_d$; $q_{out}(t)$ is the downstream outflow deviation; $q_{offtake}(t)$ is the offtake flow deviation; and $A_s$ is the average storage area. The characteristics $t_d$ and $A_s$ can be calculated by applying the system identification technique [22,23]. In fact, under different flow conditions, the values of $t_d$ and $A_s$ obtained by the system identification technique vary because of changes in the water surface of the pool. The values obtained under the initial flow conditions are assigned to $t_d$ and $A_s$.

The discrete form of Equation (7) is used to implement the control model:

$$e(k + 1) = e(k) + \frac{T_s}{A_s}\left\{q_{in}(k - k_d) - \left[q_{out}(k) + q_{offtake}(k)\right]\right\} \tag{8}$$

where $T_s$ is the control time interval and $k_d$ is the delay step, $k_d = t_d/T_s$.

Taking the difference between subsequent steps of Equation (8), we obtain

$$\Delta e(k+1) = \Delta e(k) + \frac{T_s}{A_s}\Delta q_{in}(k-k_d) - \frac{T_s}{A_s}\left[\Delta q_{out}(k) + \Delta q_{offtake}(k)\right] \tag{9}$$

where $\Delta e(k) = e(k) - e(k-1)$; and $\Delta q_{in}(k-k_d)$, $\Delta q_{out}(k)$, and $\Delta q_{offtake}(k)$ are the changes in $q_{in}(k-k_d)$, $q_{out}(k)$, and $q_{offtake}(k)$, respectively, between subsequent steps. Assuming that the delay step $k_d$ of a pool is 2, Equations (8) and (9) can be combined to form the following matrix equation:

$$\begin{bmatrix} e(k+1) \\ \Delta e(k+1) \\ \Delta q_{in}(k) \\ \Delta q_{in}(k-1) \\ \Delta q_{in}(k-2) \end{bmatrix} = \begin{bmatrix} 1 & 1 & 0 & 0 & -\frac{T_s}{A_s} \\ 0 & 1 & 0 & 0 & -\frac{T_s}{A_s} \\ 0 & 0 & 0 & 0 & 0 \\ 0 & 0 & 1 & 0 & 0 \\ 0 & 0 & 0 & 1 & 0 \end{bmatrix} \begin{bmatrix} e(k) \\ \Delta e(k) \\ \Delta q_{in}(k-1) \\ \Delta q_{in}(k-2) \\ \Delta q_{in}(k-3) \end{bmatrix} + \begin{bmatrix} 0 \\ 0 \\ 1 \\ 0 \\ 0 \end{bmatrix}[\Delta q_{in}(k)] + \begin{bmatrix} -\frac{T_s}{A_s} \\ -\frac{T_s}{A_s} \\ 0 \\ 0 \\ 0 \end{bmatrix}\left[\Delta q_{offtake}(k) + \Delta q_{out}(k)\right] \tag{10}$$

In the process of water level control, there are often water level constraints at the control points. As the water level constraints are output constraints, mandatory constraints will result in the control system having no valid solutions. Here, flexible constraints [24] are used to deal with the water level constraint problem. Let the water level deviation exceeding the constraint water level be $e^*$ and the overall water level deviation be $e$. If the upper and lower values of the water level deviation limit are $u_{min}$ and $u_{max}$, respectively, we can write

$$e^*(k) = e(k) - u^*(k) \tag{11}$$

$$u^*(k) = \begin{cases} e & u_{min} < e \leq u_{max} \\ u_{max} & e \geq u_{max} \\ u_{min} & e \leq u_{min} \end{cases} \tag{12}$$

where $u^*(k)$ is an artificial parameter. It is assumed that the water level is allowed to be within 0.35 m of the design water level. As the treatment of the water level deviation constraint is handled as a flexible constraint, the value of $u_{max}$ should be less than 0.35 m. Thus, $u_{max}$ is set to 0.25 m and $u_{min}$ is set to −0.25 m. Using these expressions for $e^*(k)$ and $u^*(k)$, Equation (10) can be rewritten as

$$\begin{bmatrix} e(k+1) \\ \Delta e(k+1) \\ \Delta q_{in}(k) \\ \Delta q_{in}(k-1) \\ \Delta q_{in}(k-2) \\ e^*(k+1) \end{bmatrix} = \begin{bmatrix} 1 & 1 & 0 & 0 & -\frac{T_s}{A_s} & 0 \\ 0 & 1 & 0 & 0 & -\frac{T_s}{A_s} & 0 \\ 0 & 0 & 0 & 0 & 0 & 0 \\ 0 & 0 & 1 & 0 & 0 & 0 \\ 0 & 0 & 0 & 1 & 0 & 0 \\ 1 & 1 & 0 & 0 & -\frac{T_s}{A_s} & 0 \end{bmatrix} \begin{bmatrix} e(k) \\ \Delta e(k) \\ \Delta q_{in}(k-1) \\ \Delta q_{in}(k-2) \\ \Delta q_{in}(k-3) \\ e^*(k) \end{bmatrix} + \begin{bmatrix} 0 & 0 \\ 0 & 0 \\ 1 & 0 \\ 0 & 0 \\ 0 & 0 \\ 0 & -1 \end{bmatrix} \begin{bmatrix} \Delta q_{in}(k) \\ u^*(k) \end{bmatrix} + \begin{bmatrix} -\frac{T_s}{A_s} \\ -\frac{T_s}{A_s} \\ 0 \\ 0 \\ 0 \\ 0 \end{bmatrix}\left[\Delta q_{offtake}(k) + \Delta q_{out}(k)\right] \tag{13}$$

Equation (13) has the following form:

$$\mathbf{x}(k+1) = \mathbf{A}\mathbf{x}(k) + \mathbf{B}\mathbf{u}(k) + \mathbf{D}\mathbf{d}(k) \tag{14}$$

where $\mathbf{A}$ represents the system matrix, $\mathbf{B}$ is the input to state matrix, $\mathbf{D}$ is the disturbance to state matrix, $\mathbf{x}$ is the state vector, $\mathbf{u}$ is the input vector, and $\mathbf{d}$ is the disturbance vector. As $\mathbf{x}(k) = \begin{bmatrix} e(k) & \Delta e(k) & \Delta q_{in}(k-1) & \Delta q_{in}(k-2) & \Delta q_{in}(k-3) & e^*(k) \end{bmatrix}^T$ and the control objectives are only $e(k)$ and $e^*(k)$, the state vector can be converted to an output vector containing only the water level deviations $e(k)$ and $e^*(k)$ through the following equation:

$$\mathbf{y}(k) = \mathbf{C}\mathbf{x}(k) \tag{15}$$

where $\mathbf{C}$ represents the state to output matrix and $\mathbf{y}$ represents the output variables.

Equations (14) and (15) are the control models in the form of state space equations for a single pool. The state space equations for a multi-pool canal can also be written in the form of Equations (14) and (15).

The ID model describes the relationship between the water level deviation and the flow deviation of a pool, but cannot reflect the water wave characteristics of the pool. When the canal pool is completely in the backwater area, the water waves hardly deform. Such waves would accumulate, causing a standing wave [15]. At this time, the control effect of the algorithm based on the ID model would be worse, as resonance is prone to occurring in canal pools. When the canal pool has some component of uniform flow (where the depth of flow is practically at a normal depth), the waves deform considerably faster than in backwater parts. Generally, it can be judged whether a pond is completely in the backwater area by comparing the actual upstream water depth of the pool and the uniform flow depth of the pool. If the upstream water depth of the pool is close to the uniform flow depth, there is a uniform flow area in the pool and resonance is unlikely to occur. If the pool depth is significantly different from the uniform flow depth, resonance is more likely [25]. The normal water depth of an open channel can be calculated using a uniform flow calculation formula. The characteristic parameters of the ID model, upstream water depths, and uniform flow depths of the 13 pools are listed in Table 2.

**Table 2.** Parameters of the integral delay (ID) model, upstream water depths, and uniform water depths of 13 pools.

| Parameter | Pool | | | | | | | | | | | | |
|---|---|---|---|---|---|---|---|---|---|---|---|---|---|
| | 1 | 2 | 3 | 4 | 5 | 6 | 7 | 8 | 9 | 10 | 11 | 12 | 13 |
| $A_s$ | 252,101 | 759,494 | 491,803 | 689,655 | 327,869 | 413,793 | 480,000 | 659,340 | 411,176 | 327,869 | 416,667 | 361,446 | 431,655 |
| $t_d$ | 15 | 48 | 33 | 46 | 21 | 60 | 31 | 70 | 24 | 35 | 57 | 41 | 75 |
| $H_{up}$ | 5.06 | 6.71 | 4.44 | 4.55 | 6.34 | 4.31 | 3.86 | 3.18 | 4.29 | 3.81 | 3.83 | 3.82 | 3.56 |
| $H_u$ | 4.71 | 4.33 | 4.16 | 3.78 | 4.62 | 4.01 | 3.25 | 3.00 | 3.46 | 3.45 | 3.68 | 3.38 | 3.35 |

Table 2 indicates that all pools in the MRP have a large average storage area $A_s$ and a long delay time $t_d$, and all the pools are in the backwater area. To prevent resonance in the pools as much as possible, a low-pass filter is used to process the real-time water level information. The high-frequency water level signals are filtered to prevent the high-frequency water level signals from being fed back. The low-pass filter performs water level processing as follows:

$$e_{FJ}(k) = F_c e_{FJ}(k-1) + (1 - F_c)e(k) \tag{16}$$

where $e_{FJ}(k)$ is the water level deviation after filtering at time $k$ and $F_c$ is the filter constant. For resonant systems, the filter time constant can be calculated using the following formula:

$$T_f = \sqrt{\frac{AR_p}{\omega_r}} \tag{17}$$

where $R_p$ is the formant height and $\omega_r = 2\pi/T_r$ is the resonant frequency. $T_r$ is the resonance period, which can be estimated as the time required for water waves to go upstream and downstream of the pool:

$$T_r = L\left(\frac{1}{v+c} + \frac{1}{c-v}\right) \tag{18}$$

where $L$ is the length of the pool; $v$ is the flow rate; and $c$ is the wave velocity.

The control time interval is set to 10 min here. As the delay times of the 13 pools of the studied area are 15, 48, 33, 46, 21, 60, 31, 70, 24, 35, 57, 41, and 75 min, the delay time steps of the 13 pools are 2, 5, 4, 5, 2, 6, 3, 7, 3, 4, 6, 4, and 8, respectively. The system matrix **A** of the linear control model of the studied area of the MRP is a $98 \times 98$ constant, the input to state matrix **B** is a $98 \times 26$ constant, the disturbance to state matrix **D** is a $98 \times 13$ constant. The state to output matrix **C** is a $26 \times 98$ constant.

The state vector $\mathbf{x}(k)$ is a $98 \times 1$ vector. If we denote the inflow deviation, delay step, water level deviation, and the virtual parameters $e^*$ and $u^*$ of pool $i$ as $q_{c,i}$, $k_{d,i}$, $e_i$, $e_i^*$ and $u_i^*$, then $\mathbf{x}(k)$ can be expressed as

$$\mathbf{x}(k) = \begin{bmatrix} \mathbf{x}_1(k) & \mathbf{x}_2(k) & \cdots & \mathbf{x}_{12}(k) & \mathbf{x}_{13}(k) \end{bmatrix}^T \tag{19}$$

where

$$\mathbf{x}_i(k) = \begin{bmatrix} e_i(k) & \Delta e_i(k) & \Delta q_{c,i}(k-1) & \cdots & \Delta q_{c,i}(k - k_{d,i}) & e_i^*(k) \end{bmatrix}^T \tag{20}$$

The input vector $\mathbf{u}(k)$ is a $26 \times 1$ vector, which can be expressed as

$$\mathbf{u}(k) = \begin{bmatrix} \Delta q_{c,1}(k) & u_1{}^*(k) & \Delta q_{c,2}(k) & u_2{}^*(k) & \cdots & \Delta q_{c,12}(k) & u_{12}{}^*(k) & \Delta q_{c,13}(k) & u_{13}{}^*(k) \end{bmatrix}^T \tag{21}$$

The disturbance vector $\mathbf{d}(k)$ is a $13 \times 1$ vector. If we denote the offtake flow deviation $q_{offtake}$ of pool $i$ as $q_{offtake,i}$, then $\mathbf{d}(k)$ can be expressed as

$$\mathbf{d}(k) = \begin{bmatrix} \Delta q_{offtake,1}(k) & \Delta q_{offtake,2}(k) & \cdots & \Delta q_{offtake,12}(k) & \Delta q_{offtake,13}(k) \end{bmatrix}^T \tag{22}$$

The output vector $\mathbf{y}(k)$ is a $13 \times 1$ vector, which can be expressed as

$$\mathbf{y}(k) = \begin{bmatrix} e_1(k) & e_2(k) & \cdots & e_{12}(k) & e_{13}(k) \end{bmatrix}^T \tag{23}$$

### 2.4. MPC Algorithm

The MPC strategy explicitly uses a simplified process model of the real system to determine control actions by minimizing an objective function. The aforementioned simplified control model with Equations (14) and (15) can be used as the process model.

An output prediction is given by the process model. In the prediction process, the predicted output of the system, $\mathbf{y}(k+i|k)$, is determined from the current state vector $\mathbf{x}(k)$ and the future control actions $\mathbf{u}(k+i|k)$. The counter $i$ indicates the number of time steps into the future. In $\mathbf{y}(k+i|k)$, the counter $i$ ranges from 1 to $p$, the prediction horizon; in $\mathbf{u}(k+i|k)$, the counter $i$ ranges from 1 to $m$, the control horizon, which should be less than or equal to $p$. It is assumed that there are control actions from the current time step $k$ to future time step $k + m$ and no control actions from time step $k + m + 1$ to future time step $k + p$.

The predicted value for the output vector one time step into the future is expressed as

$$\mathbf{y}(k+1|k) = \mathbf{C}\mathbf{x}(k+1|k) = \mathbf{C}[\mathbf{A}\mathbf{x}(k) + \mathbf{B}\mathbf{u}(k) + \mathbf{D}\mathbf{d}(k)] \tag{24}$$

With Equations (14) and (15), the prediction for the output vector two time steps into the future is

$$\begin{aligned} \mathbf{y}(k+2|k) &= \mathbf{C}\mathbf{x}(k+2|k) \\ &= \mathbf{C}\left[\mathbf{A}^2\mathbf{x}(k) + \mathbf{A}\mathbf{B}\mathbf{u}(k) + \mathbf{B}\mathbf{u}(k+1) + \mathbf{A}\mathbf{D}\mathbf{d}(k) + \mathbf{D}\mathbf{d}(k+1)\right] \end{aligned} \tag{25}$$

This process continues until the end of the control horizon is reached, where the output state vector is

$$\mathbf{y}(k+m|k) = \mathbf{C}\mathbf{x}(k+m|k) = \mathbf{C}\left[\mathbf{A}^m\mathbf{x}(k) + \sum_{i=1}^{m} \mathbf{A}^{m-i}\mathbf{B}\mathbf{u}(k+i-1) + \sum_{j=1}^{m} \mathbf{A}^{m-j}\mathbf{D}\mathbf{d}(k+j-1)\right] \tag{26}$$

After the control horizon has passed, the remaining output predictions are based on the free response only. At the end of the prediction horizon, the predicted output state vector is

$$\mathbf{y}(k+p|k) = \mathbf{C}\mathbf{x}(k+p|k) = \mathbf{C}\left[\mathbf{A}^p\mathbf{x}(k) + \sum_{i=1}^{m}\mathbf{A}^{p-i}\mathbf{B}\mathbf{u}(k+i-1) + \sum_{j=1}^{p}\mathbf{A}^{p-j}\mathbf{D}\mathbf{d}(k+j-1)\right] \quad (27)$$

The objective function, which is typically a combination of errors of output variables between a given reference and control actions over the prediction horizon, is minimized by adjusting future control actions. It can be assumed that the output reference for $e(k)$ and $e^*(k)$ is to be zero; hence, the objective function can be expressed as

$$\min_{\mathbf{u}(k)} J = \sum_{j=0}^{p}\left(\mathbf{y}^T(k+j|k)\mathbf{Q}\mathbf{y}(k+j|k)\right) + \sum_{j=0}^{m-1}\left(\mathbf{u}^T(k+j|k)\mathbf{R}\mathbf{u}(k+j|k)\right) \quad (28)$$

where $\mathbf{Q}$ is the weighting matrix of the output and $\mathbf{R}$ is the weighting matrix of the input. The problem can be summarized as minimizing the objective function by adjusting the future control actions $\mathbf{u}(k)$. However, only the first set of control actions is implemented on the canal system. The system is then updated and the process repeated. This is the rolling optimization strategy of MPC.

The elements of $\mathbf{Q}$ and $\mathbf{R}$ provide a trade-off between minimizing the water level deviations and minimizing flow changes. Generally, the element values in matrix $\mathbf{R}$ can be set to 1, and then the element values in matrix $\mathbf{Q}$ can be assigned. Larger values of the elements of $\mathbf{Q}$ result in more aggressive control, but a controller that is too aggressive or underdamped will tend to overshoot and oscillate. The elements in the output vector are $e(k)$ and $e^*(k)$, so the elements in $\mathbf{Q}$ are the weight coefficients corresponding to $e(k)$ and $e^*(k)$. For MPC, repeated tuning is performed via trial-and-error techniques [13]. The weight coefficients corresponding to $e(k)$ can be set to 10, and the weight coefficients of $e^*(k)$ can be set to 20 with the obtained results of lesser fluctuations and a shorter stabilization time.

The prediction horizon $p$ should be long enough to contain all of the delay time steps of the 13 pools, so the prediction horizon $p$ is set to 70. The control horizon $m$ should be less than prediction horizon $p$, and the control horizon $m$ is set to 40.

## 3. Results

Here, we use the simulation model instead of the actual project to test the effectiveness of the algorithm on the MRP. With a simulation model of the studied area of the MRP, we constructed the aforementioned test scenarios by changing the offtake flows in the simulation model. Then, the simulated water levels begin to change from the target levels, and there is a need to control the check gates to stabilize the water levels such that they reach the target levels again. With the linear control model, the prediction model or the process model in the MPC is established, and the MPC control can be used on the studied canal system. For the unknown disturbances, the offtake flow changes are not considered in the MPC; for the known disturbances, the information about the changes is considered. The simulation results for the test scenarios with MPC control are presented in Figures 2–9.

The critical water depths of the 13 pools at the initial condition are roughly 1.9, 1.4, 1.4, 1.4, 1.6, 1.3, 1.2, 1.2, 1.1, 1.2, 1.4, 1.3, and 1.3 m, respectively, which are considerably smaller than the target water depths, the smallest of which is 3.95 m. In Figures 3, 5, 7 and 9, however, the maximum water deviation from the target water depth is about 0.5 m. Therefore, it can be deduced that there is no occurrence of a critical flow in all four scenarios.

Figures 2 and 3 show the results for scenario 1. As can be seen from Figure 2, the maximum water level deviation is 0.06 m in pool 11. The water levels at the downstream end of all pools and the inflows of all pools fluctuate smoothly. Referring to the results of previous studies, in a feasible control system, it is normal for the inflows and water levels of the upstream pools to change this way

as a result of water delivery disturbances, which also reflects the rationality of the control system. If a water level deviation of 10% of the maximum water level deviation is regarded as stabilization by the control system, the canal system becomes stable 54 h after the disturbance (which occurs 10 h into the simulation). The stabilization time is much longer than the 6 h required by the MPC system for test canal 1, one of the test canals developed by the American Society of Civil Engineers (ASCE) Task Committee on Canal Automation Algorithms [13], but is close to the stabilization time of 40 h for the Arizona Canal, part of the Salt River Project [26]. This is because the stabilization time is mainly related to the delay time between the upstream-most inflow check gate and the disturbance point. The delay times between the upstream-most inflow check gate and the downstream-most disturbance point of the two projects mentioned above are 1.5 h and 6 h, respectively, whereas for the MRP considered in this paper, the delay time between the upstream-most inflow check gate and the disturbance point of scenario 1 is 7 h. In fact, the element values of the weight matrix **Q** also influence the control effect of the canal system. In our case, **Q** was optimized after several trial calculations. Therefore, when the disturbance occurs in the downstream-most pool, the stabilization time is the longest, at about 4–5 times the sum of the delay times of all pools. However, in Figure 2, although the water level in all pools is eventually stabilized, pools 1–5 exhibit a relatively obvious resonance phenomenon, causing the water level to oscillate vigorously before stabilization. The water level oscillation is most obvious in pool 1, causing the water level to fluctuate erratically. This resonance makes changes in the water level unpredictable, and may cause the pool to overflow. In Figure 3, the changes in the inflow to these pools is relatively smooth, which indicates that the resonance is mainly determined by the characteristics of the pools. The fundamental reason is that the water waves at the control point are obvious when the canal pool is completely in the backwater area, but this wave characteristic is not considered in the ID model. Schuurmans (1999) [15] used a low-pass filter to process the water level information of the pools in complete backwater areas, and then used the filtered water level information to generate control strategies. In his results, only significant variations in the offtake flow (greater than the flow in the pools) produced obvious resonance. In this study, as pools 1 and 5 are relatively short with a flat bottom slope, there are obvious water waves in these pools, resulting in resonance in pools 1–5, even in the case of a small change in the offtake flow.

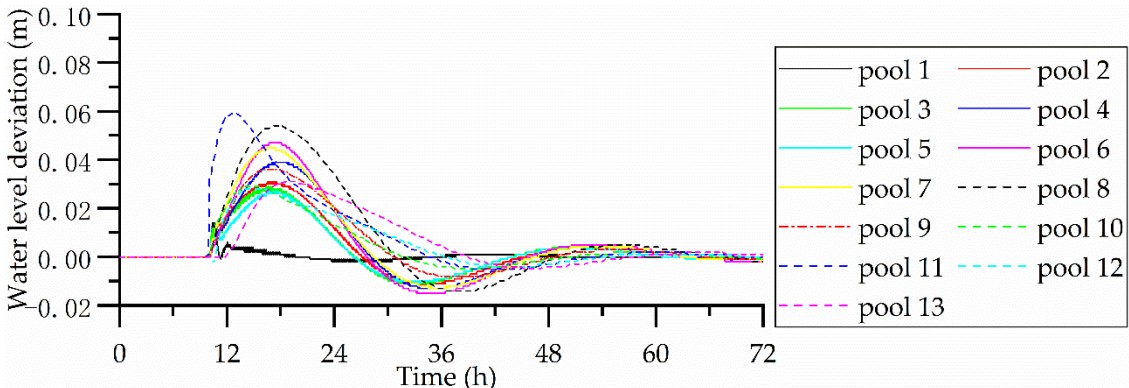

**Figure 2.** Water level deviation results for scenario 1.

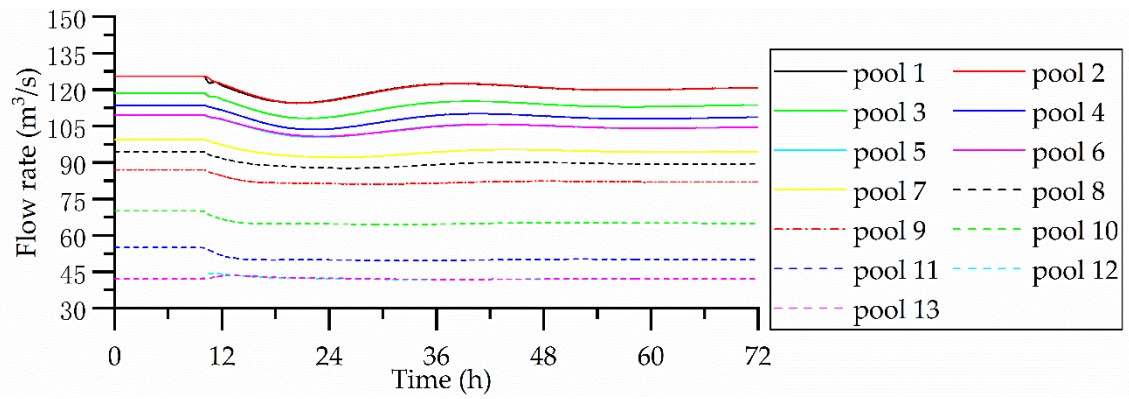

**Figure 3.** Flow rate results for scenario 1.

Figures 4 and 5 show the results for scenario 2. As this scenario considers a planned offtake change, the offtake change information is known in advance. Hence, the flows in the upstream pools can be controlled to change in advance of the offtake flow change under the MPC, thus minimizing the water level deviations. Although the offtake flow change is much greater than the offtake flow change in scenario 1, the maximum water level deviation is only 0.08 m in pool 8. As can be seen from Figure 4, the flows in pools upstream of the disturbance point start to change at the initial time, but the whole system stabilizes after about 40 h, a faster stabilization time than scenario 1. Therefore, this advance action does not significantly increase the regulation time of the canal system. In this scenario, resonance still occurs in pools 1–5.

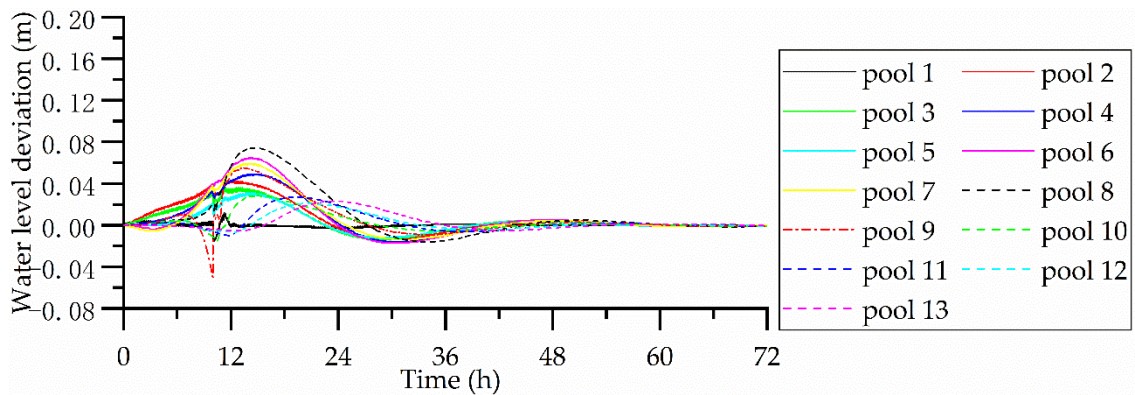

**Figure 4.** Water level deviation results for scenario 2.

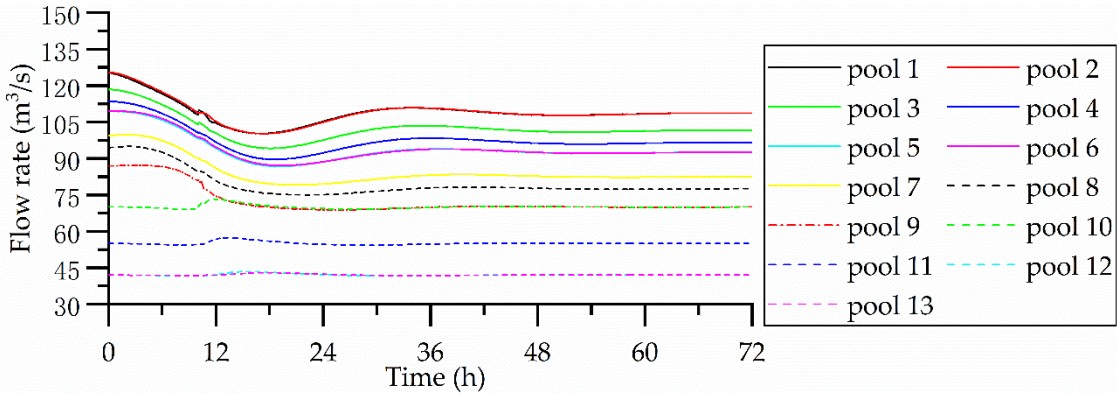

**Figure 5.** Flow rate results for scenario 2.

Figures 6 and 7 show the control results for scenario 3. The change in the offtake flow in this case is the same as in scenario 2, but the disturbance is unplanned. Compared with scenario 2, the maximum water level variation of 0.016 m is greater and the resonance phenomenon in pool 1 is more severe. The main reason is that in scenario 3, the offtake flow changes suddenly and dramatically. In the initial stage after the disturbance, the changes to the inflows of upstream pools are more frequent and violent than those in scenario 2, resulting in more obvious resonance. The changes in water level and flow rate in scenario 3 are similar to those in scenario 1, and the system requires 48 h to become stable. In scenario 3, the water diversion disturbance is relatively large, but the stabilization time does not change significantly. This shows that the stabilization time of the system is mainly related to the characteristics of the system and the control algorithm.

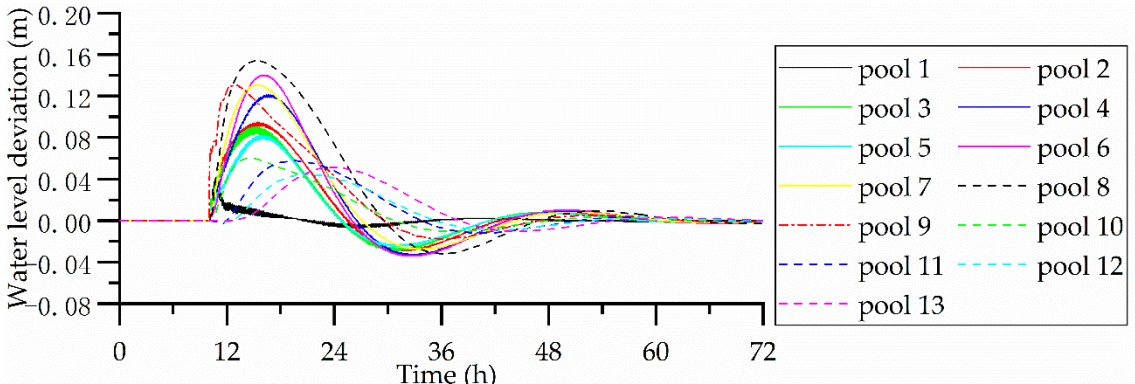

**Figure 6.** Water level deviation results for scenario 3.

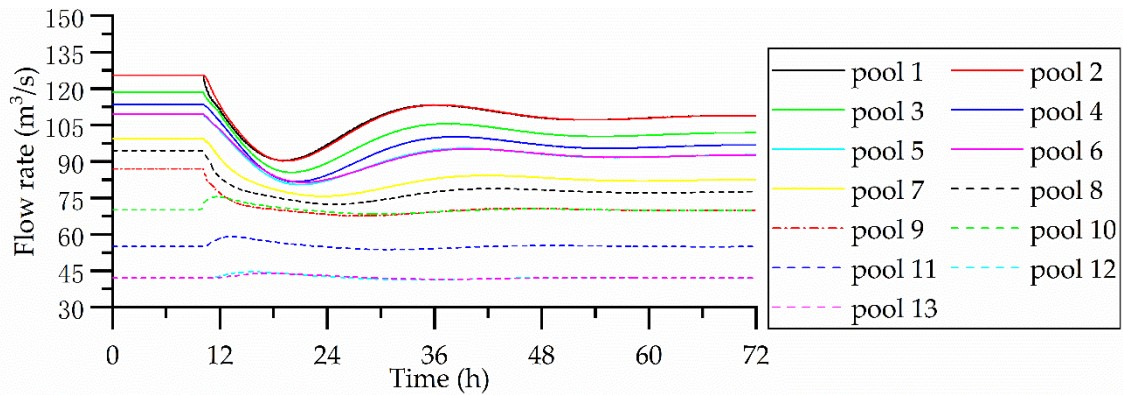

**Figure 7.** Flow rate results for scenario 3.

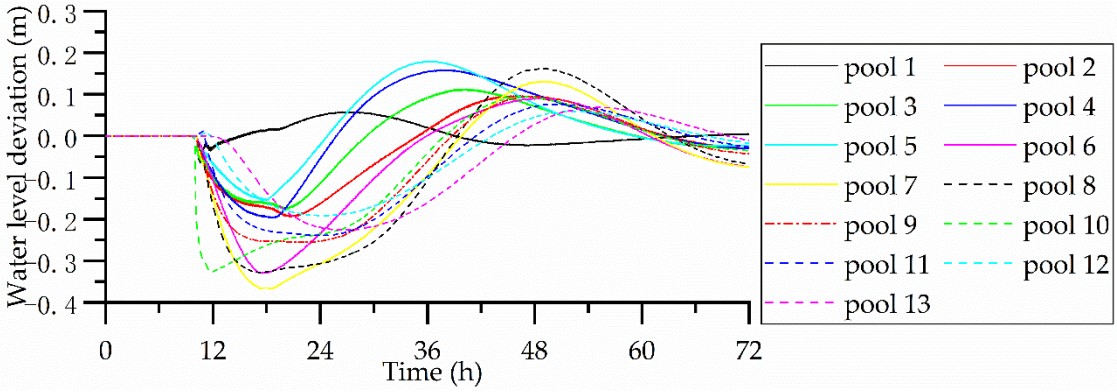

**Figure 8.** Water level deviation results for scenario 4.

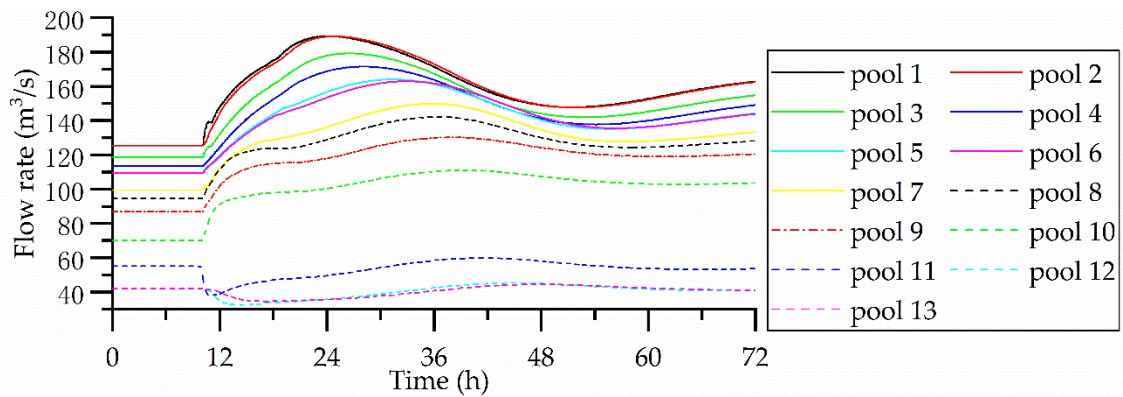

**Figure 9.** Flow rate results for scenario 4.

Figures 8 and 9 show the control results for scenario 4. As can be seen from Figure 8, following this large disturbance, the water level changes are particularly large. The greatest water level deviation is 0.37 m in pool 7, which exceeds the allowable range of water level deviation. As the water level constraint is treated as a flexible constraint in the MPC, it is not guaranteed that the water level will not exceed the allowable range when the flow rate varies significantly. Compared with the water level changes in scenarios 1 and 3, scenario 4 exhibits a somewhat different shape and a prolonged stabilization time. At the end of the calculation (72 h), the canal system had not been stabilized. Under the flexible constraints mentioned above, the weight of the water level deviation will increase when the water level deviation is too large, making the control action more intense than before and changing the trend of the water level change. However, as the characteristic parameters in the control model are obtained based on the water surface of the pools under the initial flow rate, the actual characteristics of the pools will deviate from the characteristic parameters in the control model once the flow rate changes. In the case of particularly large flow changes, the deviation between the actual characteristics of the pools and the characteristic parameters of the control model will be more pronounced, which will lead to some degradation in the control effect. However, the water level deviations of the pools gradually decrease, indicating that the canal system is stabilizing.

## 4. Discussion

The results presented in this paper show that in a reasonable canal control system, the stabilization time of the canal is mainly determined by the delay time between the upstream-most check gate and the disturbance point (stabilization then takes about 4–5 times the delay time). In the canal system studied in this research, the stabilization time is up to 40 h because the location of the disturbance is closer to the downstream end. The upstream area of the studied canal system is assumed to have a constant water level; that is, the upstream end is a reservoir. In practice, however, the pools studied here are only the 13 downstream pools in a total of 63 cascade pools in the MRP. Therefore, considering the whole set of MRP pools, when a disturbance happens at the downstream part of the canal, the stabilization time may be several days. As small disturbances are always present, a strictly stable canal system does not exist. If the disturbances are long-term positive disturbances or negative disturbances, such as when continuous intensive rainfall occurs, an excessive stabilization time will lead to the accumulation of water level deviations, which greatly increases the risk of water level overflow and limits the flexibility of changes in water demand. In terms of increasing the safety of the project, it is recommended that some reservoirs be created in the MRP to divide this long canal system into separate, shorter canal systems.

The simulations in this study used a computer with a 2.7 GHz Core i5 processor and 16 GB of RAM. With a control step of 10 min, simulating the control effect over a 72-h simulation time took 20 min, which indicates that the MPC system based on the ID model has relatively low computational complexity and can be used for real-time control. If the number of pools was extended from 13 to the

whole set of pools in the MRP, the calculation time would increase significantly. Therefore, from the perspective of shortening the calculation time of the control system, it is important to set up some reservoirs to divide the MRP into several separate canal systems.

As the model most suitable for constructing a control model for a canal system, the ID model is unable to describe the characteristics of water waves, which cannot be neglected in pools that are totally in backwater areas. The control system for such pools typically uses a low-pass filter to filter the water level signals, from which the real-time control strategy can be generated. However, the extremely flat bottom slope of the MRP and relatively short length of some pools means that the waves barely deform and would accumulate, causing a resonance effect, even for small disturbances. To build an automatic control system for the MRP, a control model that reflects the water wave characteristics can be studied. However, a complex control model will increase the computational time of the control algorithm. Also, some other advanced control methods that do not use the explicit mathematical model of the controlled process, such as model-free adaptive control (MFAC) [27] and fuzzy control [28], can be further developed to meet the application on the multipool canal system. From an engineering point of view, the resonance phenomenon caused by the application of the linear predictive control algorithm to the MRP could be eliminated by minimizing the number of short pools in the project. This could be done by combining short pools with upstream or downstream pools.

Although the control system adopted a linear process model, MPC is reasonably robust and can control the system so that it is stabilized under the condition of large disturbances. However, as the parameters of the control model are based on the initial flow, significant changes to the flow rate cause the control effect to deteriorate. Once the operating flow range has been roughly determined, the characteristic parameters of the ID model under the average operating flow can be used to construct the control model. When the disturbance is very large, even an advanced MPC algorithm cannot prevent the water levels from exceeding the allowable range. Therefore, sudden large changes in water delivery should be avoided during normal operation.

## 5. Conclusions

There are several important conclusions to this study based on the simulation results presented in this paper. These conclusions are summarized below.

1. The short length and flat bottom slope of some pools of the MRP mean that water waves barely deform in these pools. When the control system constructed based on the ID model is used for automatic control of the MRP, although the control strategy is calculated by using the filtered water level signals through the low-pass filter, resonance still occurs in some pools under conditions of small changes in water delivery.

2. In the case of a single disturbance, the stabilization time of the canal control system is approximately 4–5 times the delay time from the upstream-most check gate to the disturbance point. For the MRP, when the downstream water delivery changes, the stabilization time of the canal system will be too long. Under positive or negative disturbances over long time periods, the excessively long stabilization time may cause the water level deviations to accumulate, resulting in water level deviations that exceed the system constraints.

3. The MPC algorithm is relatively robust. Although a linear control model is used, the water level can be stabilized even under significant changes in water delivery. Although the MPC algorithm can deal with the water level constraint problem, the water level constraint is an output constraint, and so the water level cannot be guaranteed to satisfy this constraint under large changes in delivery.

4. To reduce the stabilization time of the control system and the calculation time of the MPC for the MRP, from an engineering point of view, it is suggested that some reservoirs be constructed in the MRP to divide the current canal system into several separate canal systems for regulation. Additionally, short canal pools in the system should be eliminated as much as possible.

**Author Contributions:** Conceptualization, J.Q. and L.K.; methodology, L.K.; software, L.K.; formal analysis, Q.Y.; investigation, J.Z.; resources, P.S.; data curation, P.S.; writing—original draft preparation, L.K. and J.Q.

**Funding:** This work was supported by the National Water Pollution Control and Treatment Science and Technology Major Project of the National Science and Technology Major Project of the Ministry of Science and Technology of China under Grants 2017ZX07108001.

**Acknowledgments:** The authors thank Ming Tang for their essential role in the process of data collection. We would also thank to Zhilei Zheng for giving us analysis and explanation of the use of the MPC method. The anonymous reviewers and the editor are also thanked for providing insightful and detailed reviews that greatly improved the manuscript.

**Conflicts of Interest:** The authors declare no conflict of interest.

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
