# Peer review of "Automatic Control of the Middle Route Project for South-to-North Water Transfer Based on Linear Model Predictive Control Algorithm"

_water, doi:10.3390/w11091873_

Round 1
Reviewer 1 Report
The current paper proposes to present the application of model predictive control (MPC), a popular real time control algorithm particularly suited to the automatic control of multi-pool irrigation water delivery systems, to the Middle Route Project (MRP) using a linear control model. The control system is tested in part of the MRP by means of numerical simulations.
Comments to author:
- Please add a section with the linear and nonlinear mathematical model of the Middle Route Project process.
- Please add more details of how the theory from the previous sections is applied in section 3.
- The authors should add a few words of how the parameters of the MPC were obtained/chosen.
- The state of the art can be improved with more references regarding other type of controllers, maybe the author could add the following publications:
o Second order intelligent proportional-integral fuzzy control of twin rotor aerodynamic systems, Procedia Computer Science, vol. 139, pp. 372-380, 2018.
o On model free adaptive control and its stability analysis, IEEE Transactions on Automatic Control, doi 10.1109/TAC.2019.2894586, pp. 1–14, 2019.
Author Response
Dear viewer,
Thank you for your useful comments of our manuscript. We have corrected several mistakes in our previous draft and modified the manuscript accordingly. The detailed corrections are listed below point by point. Also, we mark the important changes in red in the revised manuscript.
Point 1: Please add a section with the linear and nonlinear mathematical model of the Middle Route Project process.
Response 1: As the simulation model is a nonlinear model and equations (1)- (6) are expanded into a very large matrix equation after implicit difference transform, it will need a lot of content to describe this part and the large matrix equation is also really hard to describe, so the process to get the nonlinear mathematical model of the Middle Route Project is not in the text, we list the references about the methods, which are common methods in hydraulic modelling.
We added some contents about the linear model of the Middle Route Project process, as the matrix of the linear control model of the studied area of the MRP is a constant, the input to state matrix is a constant, the disturbance to state matrix is a constant. The state to output matrix is a constant,they are not fully expressed.
Point 2: Please add more details of how the theory from the previous sections is applied in section 3
Response 2: We now have introduced how the previous models are used in section 3 to get the simulation results.
Point 3: The authors should add a few words of how the parameters of the MPC were obtained/chosen
Response 3: We now have added a content to explain the process model of MPC is the equations (13) and (14) got in 2.3 Canal Control Model part. The reason for choose parameters Q、 R、p and m are also introduced in detail.
Point 4: The state of the art can be improved with more references regarding other type of controllers, maybe the author could add the following publications
Response 4: We now have added a sentence “Also, some other advanced control methods that do not use the explicit mathematical model of the controlled process, such as model-free adaptive control (MFAC) [27] and fuzzy control [28], can be further developed to meet the application on the multi-pool canal system.” in the Discussion part. We also hope these advanced control methods can be used in irrigation canal systems.

Reviewer 2 Report
Paper has big importance, due to the large scale of the analysed problem, as well as, and in my opinion most important-application of the scientific methods for the engineering problem in real case study. Also, complicated procedures, i.e. dynamic programming, 0-1 method, etc, could be used for such complex problems, but authors present their methodology with all details and explanations. Presented methodology will find application in similar situations. There is only need for small minor revision.
1) Authors should write comment about pressure state in the case study, due to the transport of water.
2) Figures 2-9: authors should comment if there was a occurrence of a critical flow.
3) Authors should comment about possible, even common situations which may occur, due to the intensive precipitation, when possibility of water overflow could happened.
Author Response
Dear viewer,
Thank you for your useful comments of our manuscript. We have corrected several mistakes in our previous draft and modified the manuscript accordingly. The detailed corrections are listed below point by point. Also, we mark the important changes in red in the revised manuscript.
Point 1: Authors should write comment about pressure state in the case study, due to the transport of water.
Response 1: The studied water transfer project is an open canal transfer project, so the flow is non-pressure flow and we have added this in the manuscript.
Point 2: Figures 2-9: authors should comment if there was a occurrence of a critical flow.
Response 2: We have now added a part in the Results part “The critical water depths of the 13 pools at the initial condition are roughly 1.9, 1.4, 1.4, 1.4, 1.6, 1.3, 1.2, 1.2, 1.1, 1.2, 1.4, 1.3, and 1.3 m, respectively, which are considerably smaller than the target water depths, the smallest of which is 3.95 m. In Figures 3, 5, 7, and 9, however, the maximum water deviation from target water depth is about 0.5 m. So, it can be deduced that there is no occurrence of a critical flow in all four scenarios.” As the critical water depths are considerably smaller than the initial water depths and the water depth deviations are small in figures, we can deduced that there is no occurrence of a critical flow in all four scenarios
Point 3: Authors should comment about possible, even common situations which may occur, due to the intensive precipitation, when possibility of water overflow could happen
Response 3: We here comment that when continuous intensive rainfall occurs, an excessive stabilization time will lead to the accumulation of water level deviations. It is to introduce that the risk of overflow will be high with MPC automatic control, but it is really hard to quantify the degree of the risk. As the stabilization time of the whole MRP is not quantified, and how significantly the water level will change due to a designated rainfall is not quantified. So we cannot comment about when possibility of water overflow could happen. But it can still be deduced that an excessive stabilization time will lead to the accumulation of water level deviations, which may cause overflow.